# KeyWorld: Key Frame Reasoning Enables Effective and Efficient World Models

## Abstract

Robotic world models are a promising paradigm for forecasting future environment states, yet their inference speed and the physical plausibility of generated trajectories remain critical bottlenecks, limiting their real-world applications. This stems from the redundancy of the prevailing frame-to-frame generation approach, where the model conducts costly computation on similar frames, as well as neglecting the semantic importance of key transitions. To address this inefficiency, we propose **KeyWorld**, a framework that improves text-conditioned robotic world models by concentrating transformers computation on a few semantic key frames while employing a lightweight convolutional model to fill the intermediate frames. Specifically, KeyWorld first identifies significant transitions by iteratively simplifying the robot's motion trajectories, obtaining the ground truth key frames. Then, a DiT model is trained to reason and generate these physically meaningful key frames from textual task descriptions. Finally, a lightweight interpolator efficiently reconstructs the full video by inpainting all intermediate frames. Evaluations on the LIBERO benchmark demonstrate that KeyWorld achieves a $5.68\times$ acceleration compared to the frame-to-frame generation baseline, and focusing on the motion-aware key frames further contributes to the physical validity of the generated videos, especially on complex tasks. Our approach highlights a practical path toward deploying world models in robotic control and other domains requiring both efficient and effective world models. Code is released at https://anonymous.4open.science/r/Keyworld-E43D.

## 1 Introduction

Robotic world models are generative frameworks that predict future environment states based on an initial observation and a conditioning input (Ding et al., 2024; Agarwal et al., 2025). Their ability to simulate plausible future trajectories is crucial for a variety of applications, ranging from model-based reinforcement learning (MBRL) (Luo et al., 2023; Hansen et al., 2023) to policy evaluation (Shang et al., 2025; Li et al., 2025; Kawaharazuka et al., 2024). However, the practical deployment of these models faces two significant challenges. First, their powerful predictive capability comes at a substantial computational cost, severely hindering applications like online planning. Second, the prevailing per-frame generation paradigm often fails to produce physically consistent trajectories, leading to implausible motions that undermine the utility of the simulation for downstream tasks. These bottlenecks severely limit the realism and efficiency of model-based reasoning, calling for more efficient and physically-grounded generation paradigms.

Within such robotic scenarios, observations are typically captured by a fixed camera, with the robot being the primary moving entity (Liu et al., 2023; Brohan et al., 2022). From the human perspective, it is easy to imagine the video progression by visualizing a few key motions, such as "move left", "grasp the object", and "lift". However, in stark contrast to this efficient reasoning process, current world models follow a frame-by-frame approach. They incur substantial computational redundancy by generating every frame from scratch with costly image generation modules (Wu et al., 2024; Yang et al., 2024). In addition, the standard practice of applying a uniform reconstruction loss during training forces the model to allocate its capacity equally across all frames, regardless of their semantic importance (Agarwal et al., 2025; Cen et al., 2025). This dilutes the learning signal for critical state transitions and ultimately hinders the generation of physically coherent long-horizon sequences.

An intuitive solution to reduce this redundancy is to synthesize only a sparse subset of frames (key frames) using the expensive world model, and reconstruct the remaining frames with a lightweight model conditioned on those key frames. However, each step in this roadmap is challenging: (1) **Selecting appropriate key frames.** Key frames must retain the trajectory's essential semantics while leaving intermediate motion simple enough for a lightweight interpolator. (2) **Generating key frames.** Unlike per-frame generation, this task requires the model to synthesize temporally distant anchors while preserving global coherence and physical plausibility, posing distribution-shift and long-range dependency challenges. (3) **Reconstructing between key frames.** The number of intermediate frames is unknown, and large pose differences between key frames can produce substantial motion gaps that are difficult to model.

To address the above challenges, we propose **KeyWorld**, a framework that enhances the efficiency and effectiveness of text-conditioned robotic world models through explicitly focusing the computational load on key-frame reasoning. First, we construct a motion-aware key frames dataset from robotic poses using the Ramer–Douglas–Peucker (RDP) (Ramer, 1972; Douglas & Peucker, 1973) algorithm, which retains significant motion transitions and discards the steady movements. This ensures that the preserved key frames capture essential semantics and the intervals between them remain simple enough for lightweight interpolation. With the key frame dataset, we train a Diffusion Transformer (DiT) to reason about critical motions from the task description and initial state, and then synthesize the corresponding key frames. By fine-tuning on motion-aware key frames, the model learns to generate semantically critical anchors, significantly reducing the computational burden while enhancing its focus on essential physical interactions. Finally, we employ a lightweight Convolutional Neural Network (CNN) model, which is powerful enough for reconstructing the full video sequence by predicting frame gaps and generating intermediate frames between consecutive key frames while also eliminating heavy computational overhead. We evaluate KeyWorld on the representative robotic benchmark LIBERO (Liu et al., 2023), and results demonstrate a $5\times$ acceleration compared to the frame-to-frame model. Furthermore, the motion-aware key frames guide the model to produce trajectories with higher physical plausibility, notably resulting in a substantially increased probability of the robot manipulating the correct target object. By addressing the dual challenges of efficiency and physical fidelity, our approach enables more practical deployment of world models in efficiency-sensitive robotic applications. The contribution of our work is summarized as follows:

- We propose **KeyWorld**, an efficient and modular framework that decouples text-conditioned robotic world model inference into diffusion-based key frame generation and lightweight intermediate frame interpolation. This design significantly reduces video roll-out costs and enhances semantic understanding at critical frames.

- We introduce a motion-aware key-frame detection paradigm that selects semantically critical states directly from robot pose trajectories. By aligning frame selection with meaningful physical transitions, this design not only provides a grounded abstraction of robotic videos but also fosters a sharper representation of physical dynamics within the model.

- We extensively evaluate KeyWorld on the LIBERO benchmark and demonstrate that it achieves **up to $5.68\times$ acceleration** while maintaining superior video quality across multiple metrics. These results suggest that motion-aware key-frame reasoning offers a viable path for making robotic world models more practical in time-sensitive and physics-sensitive applications

## 2 PRELIMINARIES

### 2.1 WORLD MODELS

World models aim to approximate the dynamics of an environment by forecasting its future observations. Formally, a world model is defined as a probabilistic generative model, $p(x_{1:N}|x_0, c)$, where $x \in \mathcal{X}$ is the observation image of the environment, $c \in \mathcal{C}$ is the condition, and $N$ is the length of the horizon. The conditioning variable $c$ can take different forms depending on the application. In robotic tasks, two major categories are action-conditioned and text-conditioned world models. Action-conditioned models predict future observations given the current state and detailed control inputs (Shang et al., 2025; Cen et al., 2025). In contrast, text-conditioned models generate trajectories conditioned on natural language instructions, enabling applications in high-level planning and

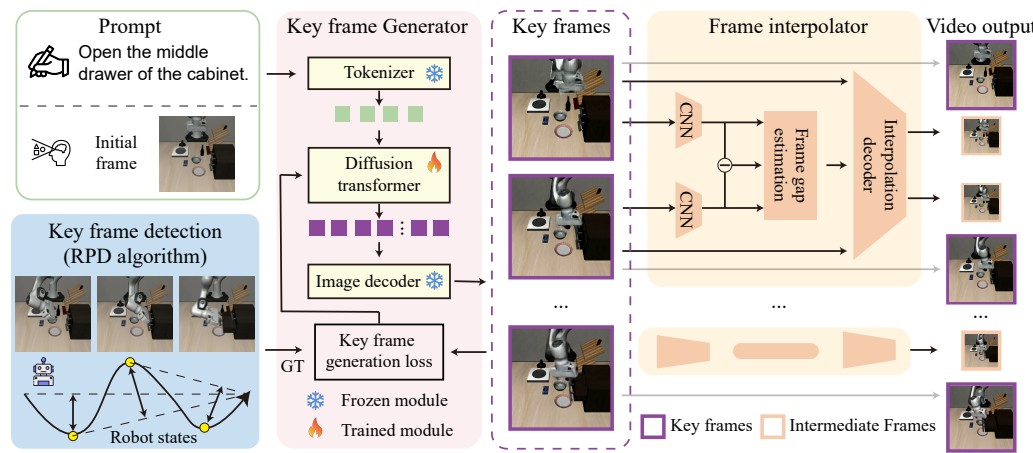

Figure 1: **The KeyWorld framework.** The pipeline comprises three stages: (1) key frame detection via the RDP algorithm, (2) key frame generation using a diffusion transformer, and (3) video reconstruction with a lightweight interpolation module.

natural language–based interaction (Zhou et al., 2024; Agarwal et al., 2025). In this work, we focus on the latter setting.

## 2.2 PROBLEM FORMULATION

We formalize the core components of the proposed KeyWorld framework as follows.

**Key Frames**. Key frames, denoted as $\{x_k \mid k \in \mathcal{K}\}$, correspond to frames that capture significant transitions between distinct robot motions. Examples include instances where the robot changes its direction of movement or switches between active joints. This motion-aware definition contrasts with conventional key frame selection strategies used in video generation or super-resolution (Arkhipkin et al., 2023), which often rely on uniform temporal sampling.

**Key Frame Generation Model**. The key frame generation model is responsible for synthesizing only the key frames conditioned on a task description and an initial observation. Formally, it learns the conditional distribution $p_\theta(x_{t\in\mathcal{K}} \mid x_0, c)$, where $c$ is a textual prompt describing the task and $x_0$ is the initial image.

**Frame Interpolation**. The frame interpolation model generates intermediate frames between consecutive key frames to reconstruct the complete video sequence. Formally, it models the distribution $p_\phi(x_{k_i+1:k_{i+1}-1} \mid x_{k_i}, x_{k_{i+1}})$, where $k_i, k_{i+1} \in \mathcal{K}$ are adjacent key frames.

**World Modeling with Key Frames**. Using the above components, we decompose the world modeling objective into two sub-tasks: key frame generation and frame interpolation. The overall distribution is approximated as:

$$p(x_{1:N} \mid x_0, c) \approx \left[ \prod_{i=1}^{|\mathcal{K}|-1} p_\phi(x_{k_i+1:k_{i+1}-1} \mid x_{k_i}, x_{k_{i+1}}) \right] p_\theta(x_\mathcal{K} \mid x_0, c) \tag{1}$$

## 3 METHODS

### 3.1 OVERVIEW

In this section, we present the detailed design of constructing the KeyWorld framework, which is illustrated in Figure 1. First, key frames are extracted from robot trajectories to construct a training dataset for the key frame generator (Section 3.2). We then train the two modules of the video-generation model: (1) a DiT key frame generator that synthesizes high-quality key frames capturing

the essential motions (Section 3.3), and (2) a CNN lightweight frame interpolation model that produces the intermediate frames to reconstruct full video sequences (Section 3.4).

## 3.2 KEY FRAME EXTRACTION

As key frames correspond to transitions in the robot's movement, they typically manifest as local extrema in the robot's pose trajectory. The motion between two consecutive key frames is relatively simple and smooth, making it tractable for a frame interpolation model to reconstruct the intermediate frames accurately. This motivates us to formalize key frame selection as a problem of trajectory simplification, which aims to preserve critical turning points while discarding redundant states. In this work, we adopt the Ramer–Douglas–Peucker (RDP) algorithm (Ramer, 1972; Douglas & Peucker, 1973) to identify key frames from robot pose vectors. The iterative procedure is summarized as follows:

$$
R(s_{0:N}) = \begin{cases} R(s_{0:i^*}) \cup R(s_{i^*:N}), & \text{if } \dfrac{d(s_{i^*}, \overline{s_0 s_N})}{\|s_N - s_0\|} \geq \epsilon, \\ \{s_0, s_N\}, & \text{otherwise}, \end{cases} \qquad i^* = \arg \max_{1 \leq i \leq N-1} d(s_i, \overline{s_0 s_N}). \quad (2)
$$

Here, $s_i \in \mathcal{S}$ denotes the robot's state at step $i$, such as end-effector positions and joint angles. $d(s_i, \overline{s_0 s_N})$ represents the distance from $s_i$ to the line segment $\overline{s_0 s_N}$. Intuitively, the algorithm recursively selects the state point that deviates the most from the chord connecting the current segment endpoints, retaining only those points that exceed a threshold $\epsilon$. This ensures that the selected key frames capture the most significant kinematic changes in the trajectory. We conduct a binary search on the threshold $\epsilon$ to control $|\mathcal{K}|$, the number of key frames.

## 3.3 KEY FRAME GENERATION MODEL

To synthesize high-quality key frames conditioned on both the initial state and the textual task description, we require a powerful conditional video generation model. We adopt CogVideoX (Yang et al., 2024), a state-of-the-art diffusion-based image-to-video model, as the backbone of our key frame generator. We specifically finetune the model on key-frame–organized video subsequences, encouraging it to capture the physically and semantically meaningful aspects of the robot-environment interaction. By focusing on these informative states, the model learns to reason on significant transitions in the trajectory, which provides a stronger foundation for subsequent frame interpolation and ensures that reconstructed sequences maintain coherent dynamics.

Importantly, since CogVideoX is pretrained on general video data rather than the robotics domain, finetuning is indispensable for effective deployment in our setting. This finetuning step is a prerequisite for any method employing such a base model in robotics, whether it generates all frames or only key frames. Thus, it constitutes a common overhead and does not introduce additional training cost unique to our key frame-based approach. Considering the autoregressive nature of transformer blocks in the diffusion transformer (DiT) module, the computational cost scales linearly with the sequence length at both training and inference stages. As a result, generating only the key frames reduces the cost to approximately $\frac{|\mathcal{K}|}{N}$ of that required for synthesizing all $N$ frames.

## 3.4 RECONSTRUCTING KEY FRAMES TO FULL VIDEO

Having generated the sparse set of key frames, the subsequent step is to reconstruct the complete, temporally smooth video sequence. We design a lightweight frame interpolation model for this task, which consists of two dedicated components: (1) a gap estimator that predicts the number of intermediate frames between consecutive key frames, and (2) a frame interpolator that synthesizes the corresponding frames accordingly.

**The gap estimator** predicts the gap $g_i$ between consecutive key frames $x_{k_i}, x_{k_{i+1}}$, which is a typical regression task. Since the robot generally moves at a relatively stable speed, the difference in robot poses in frames can serve as a proxy for estimating the temporal gap between frames. Specifically, each key frame is first encoded into a latent representation using a pre-trained Convolutional Neural Network (CNN). To explicitly capture the inter-frame differences, we augment the representation by

concatenating both individual frame embeddings and their difference, yielding

$$z = \{Enc(x_{k_i}) \oplus Enc(x_{k_{i+1}}) \oplus Enc(x_{k_{i+1}}) - Enc(x_{k_i})\}, \qquad (3)$$

where $Enc(\cdot)$ denotes the CNN encoder and $\oplus$ represents the concatenation operator. The resulting feature $z$ is then passed through a Multilayer Perceptron (MLP) regression head to predict the gap. To control the length of the generated videos, we normalize the gap values at the trajectory level and apply truncation. The gap estimator is then trained using a Mean Squared Error (MSE) loss.

**The frame interpolator** is responsible for generating $g_i$ intermediate frames between consecutive key frames $x_{k_i}$ and $x_{k_{i+1}}$, which naturally fits the standard frame interpolation task. To this end, we adopt FILM (Reda et al., 2022), a CNN-based model specifically designed for high-quality interpolation in dynamic scenes, as our base interpolator. To better adapt FILM to the robotics domain and high-resolution videos, we finetune it on the key frame dataset constructed in Section 3.2.

Both modules are designed to be computationally efficient. The gap estimator contains roughly 25M parameters, while the frame interpolator has fewer than 35M. Due to the relatively simple motion between key frames, the lightweight interpolator can accurately reconstruct the full video without requiring expensive computations. In practice, converting key frames to a complete video takes less than 20 seconds, accounting for only 5% of the total pipeline runtime. Overall, the computational cost of the frame interpolation module is negligible, and the total pipeline resource usage is roughly equivalent to that of generating the key frames, which is approximately $\frac{|\mathcal{K}|}{N}$ of generating all frames with the large key frame generation model.

## 4 EXPERIMENTS

### 4.1 EXPERIMENTAL SETTINGS

**Datasets**. We conduct training and evaluation using the LIBERO dataset (Liu et al., 2023), which provides diverse scenarios for robotic arm control with rich semantic variations. The datasets include textual descriptions, videos, and robot actions in each episode, making it well-suited for building world models. The benchmark is organized into five subsets: LIBERO-90, LIBERO-10, LIBERO-goal, LIBERO-spatial, and LIBERO-object. Among them, LIBERO-goal, LIBERO-spatial, and LIBERO-object focus on testing specific generalization abilities, such as goal specification, spatial reasoning, and object variation, while the other two contain more general tasks. In all experiments, we train our model on LIBERO-90 and evaluate on the remaining four subsets. In addition, we make pre-processing of the dataset, such as removing dummy episodes and resizing, according to the previous works (Cen et al., 2025). Around 5000 episodes are preserved, and 3559 are used for training. We further aligned the dataset to 81 frames per episode at 16 Hz frequency.

**Model Implementation**. We select only 20% of frames as key frames to train the KeyWorld model, where this density is analyzed through the elbow method (Appendix F). We compare our framework with generating all frames without interpolation, which is equivalent to treating all frames as key frames, and we denote it as **frame-to-frame**. Each component of the KeyWorld models is trained separately. For the key frame generator, we follow the supervised finetuning (SFT) recommended strategy of CogVideoX1.5-5B-I2V[1]: extracting key frames into short video slices, and only updating the transformer block parameters during training. We fine-tune the model on the entire LIBERO-90 dataset at a resolution of $768 \times 1360$ for 10 epochs, which takes about 18 hours on $8 \times$ NVIDIA A800-SXM4-40GB GPUs. For the gap estimator, we adopt ResNet-50 (He et al., 2016) as the backbone CNN. For the frame interpolator, we fine-tune the model from its official checkpoints [2]. Training these latter two components requires less than 3 hours on a single GPU. Additional details about model training can be found in Appendix A.

**Metrics**. For computational efficiency, we conduct all the inference experiments on a single NVIDIA A800-SXM4-40GB GPU and report the average inference time of each model to compare computational efficiency.

For video quality, we evaluate generated videos with the following metrics:

---

[1] https://huggingface.co/zai-org/CogVideoX1.5-5B-I2V
[2] https://github.com/google-research/frame-interpolation

Table 1: Inference time (seconds) of the KeyWorld and frame-to-frame framework. The frame-to-frame model does not involve interpolation.

| Dataset | KeyWorld | | | | Frame-to-frame |
|---|---|---|---|---|---|
| | Key frame generation | Gap estimation | Frame interpolation | Total | Frame generation |
| LIBERO-10 | 160.40 | 0.35 | 11.97 | 172.72 | 1001.54 |
| LIBERO-goal | 161.00 | 0.22 | 11.74 | 172.96 | 987.98 |
| LIBERO-object | 158.74 | 0.23 | 11.82 | 170.79 | 963.70 |
| LIBERO-spatial | 158.01 | 0.28 | 11.65 | 169.94 | 1011.17 |

Table 2: Video generation quality of the KeyWorld and the frame-to-frame framework. A higher PSNR, SSIM, and object-level accuracy represent a better result, which is marked with **bold**.

| Dataset | KeyWorld | | | Frame-to-frame | | |
|---|---|---|---|---|---|---|
| | PSNR | SSIM | Object-level accuracy | PSNR | SSIM | Object-level accuracy |
| LIBERO-10 | 19.07 | **0.8750** | **90%** | **19.46** | 0.8153 | **90%** |
| LIBERO-goal | **22.43** | **0.8838** | **90%** | 20.69 | 0.8719 | 38% |
| LIBERO-object | **22.23** | **0.8801** | 62% | 21.22 | 0.8579 | **67%** |
| LIBERO-spatial | **21.40** | **0.8694** | **33%** | 19.95 | 0.8579 | 27% |

- **PSNR**. A pixel-level metric that measures the similarity between predicted and ground-truth frames based on mean squared error.

- **SSIM**. A perceptual metric that evaluates structural similarity, capturing luminance, contrast, and texture differences between images.

- **Object-level accuracy**. We manually check 20% of the trajectories, which forms a total of 100 in each subset. We report the ratio of the robot operating on the proper object.

## 4.2 EFFICIENCY AND EFFECTIVENESS

The inference time of different models is summarized in Table 1. Compared with the frame-to-frame model, KeyWorld significantly reduces the overall latency, taking less than 25% of that of the frame-to-frame model The vast majority of the computational cost arises from the key frame generator (over 90%), while the gap prediction module introduces negligible overhead, and the frame interpolation module accounts for only a minor portion. This confirms that our decomposition strategy effectively concentrates the computation on key frames and substantially reduces the overhead for non-key frames.

We also find that focusing on key frames makes the diffusion key frame generator more attentive to the physical semantics of the video. As shown in Table 2, using motion-aware key frames significantly improves the probability that the robot correctly identifies the object, while maintaining good pixel-level fidelity. This is likely because decoupling the generation process allows the key frame model to focus exclusively on high-level physical semantics, freed from the dual burden of maintaining low-level temporal smoothness and pixel-wise fidelity between every consecutive frame. We further provide a representative visualization of the LIBERO-goal subset in Figure 2, where the KeyWorld framework significantly improves object-level accuracy. The results demonstrate that our model accurately identifies the target object (the wine bottle) and generates robot motions that closely resemble the ground truth, while the frame-to-frame model failed (red circles). We attach more visualization in Appendix C.

We further validate the effectiveness of individual components, showing that our motion-aware key frames capture meaningful physical semantics (Appendix D), and the lightweight gap predictor operates with high accuracy (Appendix E).

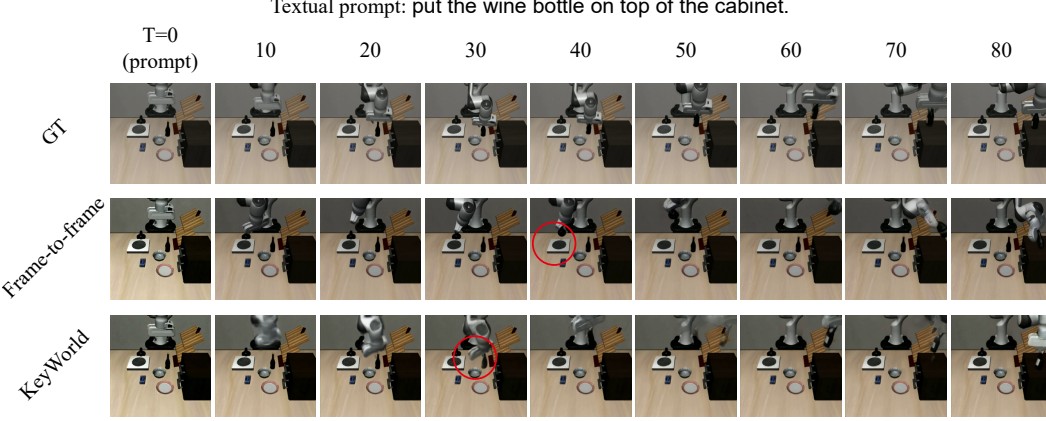

Figure 2: Example of the generated video. GT refers to the ground truth. Frames are selected uniformly from the video, regardless of whether they are key frames.

### 4.3 EFFECTIVENESS OF THE KEY FRAMES

Table 3: Video generation quality of different key frame arrangements. **Bold** represents better results between KeyWorld and Uniform models.

| Model | KeyWorld | | | Uniform | | | Visual | | |
|---|---|---|---|---|---|---|---|---|---|
| Metrics | PSNR | SSIM | Object-level accuracy | PSNR | SSIM | Object-level accuracy | PSNR | SSIM | Object-level accuracy |
| LIBERO-10 | **19.07** | **0.8750** | **90%** | 18.66 | 0.8400 | 88% | 18.94 | 0.8738 | 87% |
| LIBERO-goal | **22.43** | **0.8838** | **90%** | 21.97 | 0.8733 | 86% | 22.11 | 0.8830 | 89% |
| LIBERO-object | **22.23** | 0.8801 | 62% | 22.13 | 0.8844 | **70%** | 22.01 | **0.8868** | 59% |
| LIBERO-spatial | **21.40** | 0.8694 | **33%** | 21.32 | 0.8563 | 27% | 21.21 | **0.8700** | 31% |

We evaluate the effectiveness of extracting key frames from robot states. In particular, we compare KeyWorld with a baseline that allocates the same number of key frames uniformly across the sequence (**Uniform**). We also adopt a pure vision-based method that uses ResNet He et al. (2016) to capture the visual embedding, which is then dimension reduced by PCA as surrogate robot states (**Visual**). As shown in Table 3, KeyWorld-20 produces higher-quality videos. The improvement stems from the key frame generator producing key frames that yield simpler motions between consecutive key frames, which the lightweight frame interpolation module can model more effectively. Importantly, these three methods incur nearly identical computational costs; the only distinction is that Uniform omits the gap prediction stage, whose overhead is less than 1% of the total runtime.

We provide a detailed frame-by-frame comparison among the three key frame arrangements in Figure 3. The key frames selected by our method exhibit clear physical significance, accurately capturing critical state transitions such as the arm moving left (frames 72–74) and releasing the can (frames 74–80). In contrast, the uniformly sampled key frames are semantically misaligned with the robot's motion, often falling within simple, continuous movements. This misalignment forces the interpolation module to infer overly large motions, resulting in pronounced visual distortions and blurring in the reconstructed video. The visual-based key frames fail to capture the detailed motions, such as the release of the end effector, which are detailed but physically critical for robot operations. This leads to desynchronized or incorrect temporal pacing relative to the ground truth.

### 4.4 ADVANTAGES IN COMPLEX TASKS

To better understand the conditions under which motion-aware key frames bring the most benefit, we conduct an analysis of the relationship between trajectory complexity and the performance improvement achieved by our method. Trajectory complexity is quantified by computing the cumulative absolute difference of robot states along the demonstration trajectory, which reflects the total amount of kinematic variation within an episode and serves as a proxy for task difficulty. Object-level accuracy is not included here as it is a discrete variable.

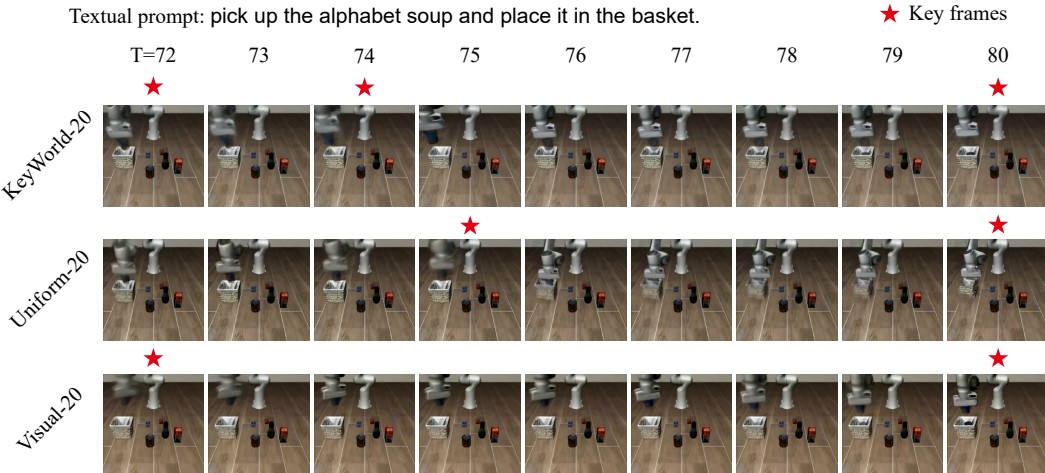

Figure 3: Example of the generated videos at different key frame arrangements. The last 9 frames of the generated video are shown.

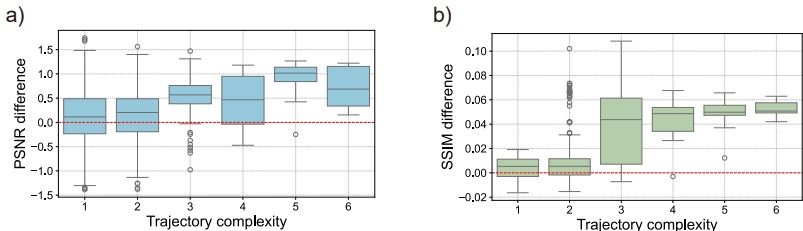

Figure 4: Performance comparison between KeyWorld-20 and Uniform-20. A positive PSNR (a) and SSIM (b) difference indicates that KeyWorld-20 performs better than the uniform baseline.

As shown in Figure 4, the performance gain from motion-aware key frames is most pronounced in scenarios with higher trajectory complexity, while tasks with relatively simple or repetitive motions see only marginal improvement. This trend indicates that our approach is particularly effective in complex scenarios where accurate modeling of significant state transitions is crucial. We attribute this effect to the fact that motion-aware key frames provide anchors that better align with physically meaningful transitions in the trajectory. By emphasizing these informative points, the generative model receives stronger guidance, which improves both the fidelity and plausibility of the reconstructed sequences. This analysis further highlights the importance of incorporating trajectory structure into key frame selection rather than relying solely on uniform sampling.

## 4.5 IMPACT OF KEY FRAME DENSITY

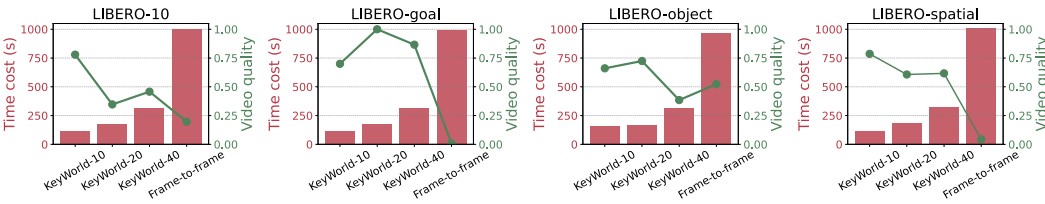

Figure 5: Computational cost (bar) and general video quality (line) of the KeyWorld variants and frame-to-frame model. A higher video quality index represents better quality.

In this section, we discuss the impact of the number of key frames, which is one of the most critical hyperparameters in the pipeline. By controlling the ratio of key frames in all frames, we train two more variants of the KeyWorld framework. Specifically, 10% and 40% of frames are selected as key frames, resulting in KeyWorld-10 and KeyWorld-40, respectively. The model evaluated in the previous section, which uses 20% key frames, is denoted as KeyWorld-20 here. We normalize the metrics to a 0-1 scale and use the coefficient of variance (CV) as a weight to report an averaged performance across different tasks. Original data used to calculate the video quality index is attached in Appendix B

Results of the four test sets are shown in Figure 5, which leads to two main conclusions: First, computational cost scales nearly linearly with the number of key frames, as their generation by the large model dominates the overall overhead. All variants achieve significant acceleration compared with frame-to-frame, and the rate of acceleration ranges from $3.12\times$(KeyWorld-40) to $8.56\times$ (KeyWorld-10). Second, using key frames consistently strengthens the video generation quality compared to the frame-by-frame baseline. While all key-frame variants demonstrate clear improvement, the 10% and 20% variants show a better performance, which is consistent to our analysis on the optimal key frame density (Appendix F).

## 5 RELATED WORKS

World models have been applied to a wide spectrum of tasks. They have been leveraged for *decision-making and policy learning*, where agents conduct planning or train latent-space actor-critic policies through imagined rollouts (Hafner et al., 2025; Li et al., 2025). Another critical line of work is *data augmentation and simulation*, where models generate synthetic yet physically plausible trajectories to enrich offline datasets and improve policy robustness and generalization (Shang et al., 2025; Wang et al., 2024; Nomura & Murata, 2023). Despite this diversity, all applications rely on repeatedly rolling out trajectories within the world model. However, the prevailing frame-to-frame generation paradigm suffers from high computational costs and often lacks physical plausibility, limiting its practical utility.

Recent advances in world models and, more broadly, video generation have increasingly leveraged key-frame-based strategies to improve efficiency and long-horizon consistency. For instance, FusionFrames (Arkhipkin et al., 2023) introduces a two-stage text-to-video pipeline, where diffusion is first applied to generate key frames that capture the overall storyline, followed by interpolation to reconstruct smooth motion between them. In the robotics domain, RoboEnvision (Yang et al., 2025) decomposes long-horizon manipulation tasks into atomic goals, generating semantically aligned key frames and then interpolating intermediate frames to enhance long-context performance. Beyond robotics, key-frame strategies have also been shown to improve coherence in long, multi-event video generation (Huang et al., 2025).

A key distinction between these works and our proposed KeyWorld model lies in the arrangement of key frames. Prior methods primarily adopt uniformly distributed key frames (Arkhipkin et al., 2023) or key frames derived from task decompositions in forms of natural language (Yang et al., 2025; Huang et al., 2025). Our method selects key frames based on physical motion transitions, which naturally minimizes the complexity of the intermediate interpolation task and leads to more physically coherent results. Thereby, our approach directly reasons about and generates only semantically critical frames, ensuring both efficiency and physical coherence.

## 6 CONCLUSIONS

In this paper, we introduced KeyWorld, a key-frame–based framework for accelerating text-conditioned world models. By concentrating expensive model computation on a sparse set of semantically critical frames and leveraging a lightweight interpolator for the remainder, KeyWorld achieves a significant $5.68\times$ acceleration on the LIBERO benchmark compared with the frame-to-frame strategy. Beyond acceleration, the motion-aware selection of these frames is key to enhancing the semantic coherence and physical validity of the generated videos. The benefits in video quality are even more significant in more complex tasks. This demonstrates a promising path toward building efficient and effective world models for robotics.

ETHICS STATEMENT

We fully use open-source models and datasets in the paper, which involve no problem regarding privacy and copyright. We cite the resources in Section 4.1. This work does not involve human subjects, discrimination, bias, or fairness concerns.

REPRODUCIBILITY STATEMENT

For Reproducibility, we describe the general experimental settings in Section 4.1; we list the implementation details in Appendix A; and our source code is anonymously open source at `https://anonymous.4open.science/r/Keyworld-E43D`.

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

# A  IMPLEMENTATION DETAILS

In this section, we provide all implementation details for reproducibility in Table 4. For fine-tuning the frame interpolator, we find it suffers from gradient explosion when training for more than 1 epoch, and it converges well on the large dataset with 1 epoch.

Table 4: Implementation details

| Module | Element | Detail |
|---|---|---|
| System | OS | Ubuntu 22.04.5 |
| | CUDA | 12.2 |
| | Python | 3.10 |
| | Pytorch | 2.6.0 |
| | Device | 8*NVIDIA A800 40G |
| Key frame generator | Batch size | 1 |
| | Number of epochs | 10 |
| | Resolution | 768*1360 |
| | Optimizer | AdamW |
| | Learning rate | 2e-5 |
| | Weight decay | 1e-4 |
| | Random seed | 42 |
| Gap estimator | Batch size | 8 |
| | Number of epochs | 100 |
| | Optimizer | Adam |
| | Learning rate | 3e-4 |
| Frame interpolator | Batch size | 4 |
| | Number of epochs | 1 |
| | Optimizer | Adam |
| | Learning rate | 5e-5 |

# B DATA USED FOR THE KEY FRAME DENSITY EXPERIMENT.

We attach the original data for Figure 5 below.

Table 5: Original data for the key frame density experiment.

| Dataset | Metric | Model | | | |
|---|---|---|---|---|---|
| | | KeyWorld-10 | KeyWorld-20 | KeyWorld-40 | Frame-to-frame |
| LIBERO-10 | PSNR | 19.13 | 19.07 | 19.19 | 19.46 |
| | SSIM | 0.8744 | 0.8750 | 0.8837 | 0.8153 |
| | Object-level accuracy | 97% | 90% | 90% | 90% |
| | Time | 115.81 | 172.72 | 316.25 | 1001.54 |
| LIBERO-goal | PSNR | 22.17 | 22.43 | 22.43 | 20.69 |
| | SSIM | 0.8708 | 0.8838 | 0.8780 | 0.8719 |
| | Object-level accuracy | 77% | 90% | 83% | 38% |
| | Time | 116.50 | 172.96 | 317.28 | 987.98 |
| LIBERO-object | PSNR | 22.00 | 22.23 | 22.12 | 21.22 |
| | SSIM | 0.8913 | 0.8801 | 0.8816 | 0.8579 |
| | Object-level accuracy | 60% | 62% | 54% | 67% |
| | Time | 114.91 | 170.79 | 314.56 | 963.70 |
| LIBERO-spatial | PSNR | 21.07 | 21.40 | 21.59 | 19.94 |
| | SSIM | 0.8522 | 0.8694 | 0.8613 | 0.8579 |
| | Object-level accuracy | 42% | 33% | 34% | 27% |
| | Time | 115.78 | 181.76 | 322.38 | 1011.17 |

## C  MORE VISUALIZATION EXAMPLES

We propose an additional visualization example in Figure 6. Results show that our KeyWorld series model generates videos similar to ground truth.

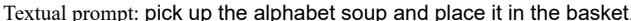

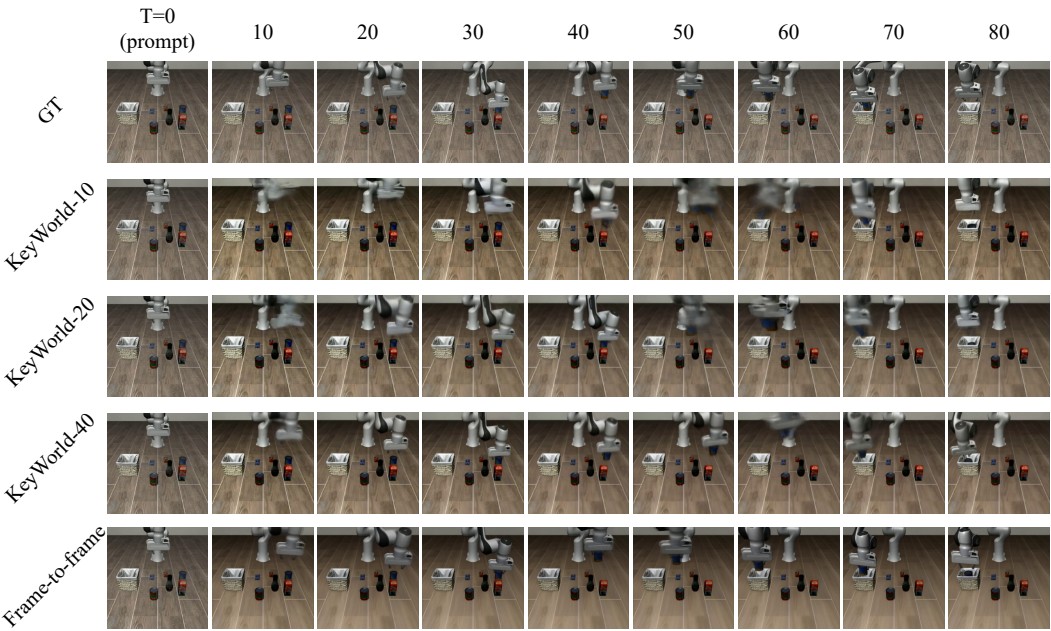

Figure 6: Example of the generated video. GT refers to the ground truth. Frames are selected uniformly from the video, regardless of whether they are key frames.

# D  EXAMPLE OF DETECTED KEY FRAMES

We illustrated the detected key frames in figure 7. We find that the key frames successfully capture the key motions of the robot. In the first example, the robot moves forward in frames 5-15 while moving downward in frames 15-25. The key frame also captures the detailed movement of the robot. For instance, they detect the movement of the end effector with high accuracy (frame 29-41 in example 1, frame 18-27 in example 2).

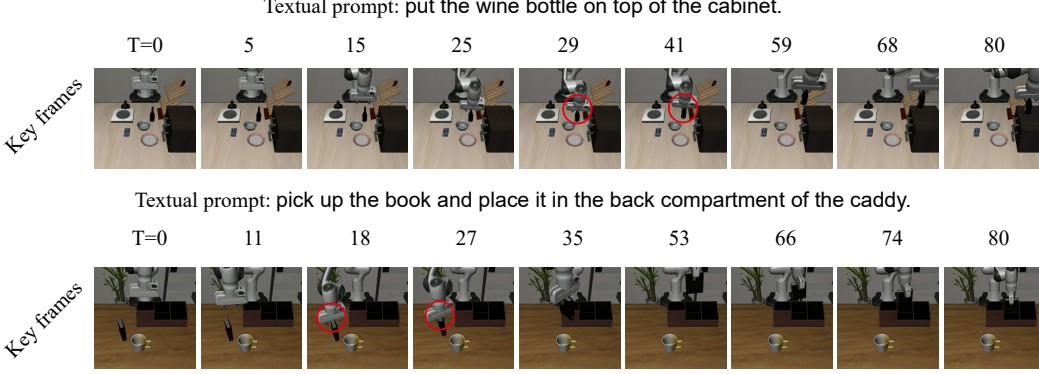

Figure 7: Example of the detected key frames on the LIBERO dataset. Videos are selected from the ground truth.

To further validate the effectiveness of motion-aware key frames, we conduct the same key frame detection strategy on the Agibot World Challenge Bu et al. (2025) dataset. The result in Figure 8 demonstrates that the motion-derived key frames align well with the semantic stages of the task, such as reaching, grasping, and lifting the object. This demonstrates that kinematic turning points are indeed strong proxies for meaningful visual and task-relevant changes across multiple datasets.

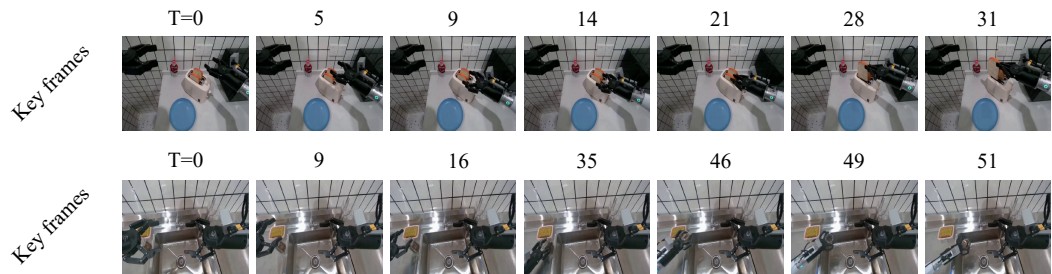

Figure 8: Example of the detected key frames on the Agibot World Challenge dataset. Videos are selected from the ground truth.

# E   ACCURACY OF THE GAP PREDICTION MODULE

The gap prediction module relies on only the consecutive key frames to predict the gap between them. Therefore, a relatively steady moving mode is assumed. We analyze moving speed on multiple datasets and check this assumption. A K-means clustering and t-SNE visualization of the speed vectors is demonstrated in Figure 9, which shows that the motions form several well-separated clusters, indicating that the robot follows a small set of motion primitives with relatively consistent speed patterns. This supports the stability assumption behind our gap estimator and suggests that it remains applicable across similar robotic manipulation domains.

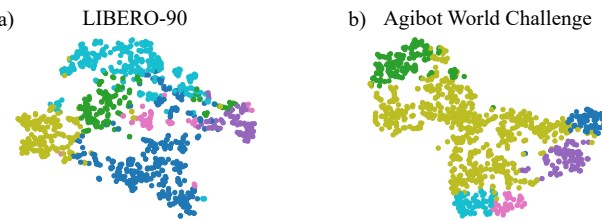

a)   LIBERO-90            b)   Agibot World Challenge

Figure 9: T-SNE visualization of the clustering result of the velocity vectors.

We report the RMSE of our gap predictor in Table 6. Overall, they achieve acceptable accuracy for frame interpolation, representing an average error of less than 0.4s under an fps of 16.

Table 6: RMSE of the gap prediction module. The error is calculated by the number of frames.

| Dataset | Model | | |
|---|---|---|---|
| | KeyWorld-10 | KeyWorld-20 | KeyWorld-40 |
| LIBERO-10 | 4.85 | 2.42 | 1.19 |
| LIBERO-goal | 5.71 | 2.46 | 1.59 |
| LIBERO-object | 5.88 | 3.57 | 2.13 |
| LIBERO-spatial | 4.99 | 2.27 | 1.39 |

# F    SELECTING THE OPTIMAL NUMBER OF KEY FRAMES.

The density of key frames is a key hyperparameter in the KeyWorld model. We propose an elbow method based on the residual curve of the RDP algorithm to seek the best density, which is analogous to choosing the number of clusters in K-means. As the key-frame density increases, the RDP residual monotonically decreases because the simplified trajectory becomes closer to the original one. We consistently observe an elbow point in this curve, indicating where the marginal gain of adding more key frames becomes negligible. This elbow reflects the intrinsic motion complexity of the sequence and therefore provides an automatic criterion for selecting the key-frame ratio. As shown in Figure 10, the elbow points of all LIBERO subsets lie around 10–15 key frames, which matches the empirical performance observed in Figure 5: KeyWorld-10 (9 key frames) and KeyWorld-20 (17 key frames) outperform other key-frame densities. This confirms that the elbow-based automatic selection aligns well with the optimal performance region.

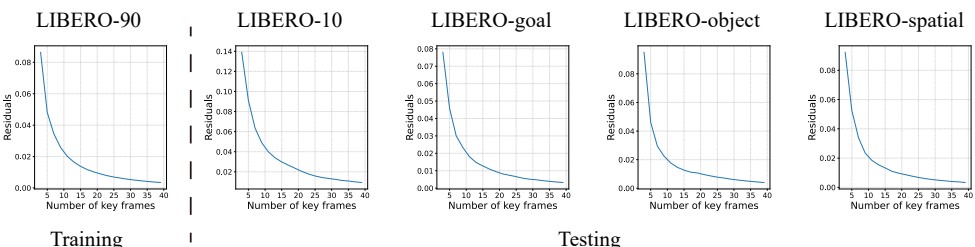

Figure 10: Residue of the RPD algorithm at different numbers of key frames.

# G  FAILURE MODES

We carefully inspected the generated videos and summarized the dominant failure modes, as shown in Figure 11.

**(1) Object mis-targeting.** In some cases, the model interacts with an incorrect object instance (e.g., opening the top drawer instead of the middle one). Our object-level evaluations show that KeyWorld significantly reduces this type of error compared with frame-by-frame baselines, but a few residual failures remain. These errors primarily originate from inaccuracies in key-frame selection. Incorporating a physics-grounded post-training stage on the base model could further mitigate this issue.

**(2) Blur and distortion during fast motion.** A second class of failures arises from motion blur or geometric distortion, particularly around fast-moving components such as the robot arm. Because the key frames themselves are visually accurate, this issue is attributed to the frame-interpolation module. When the predicted temporal gaps between key frames deviate from the true motion profile, the interpolator produces unstable motion speeds and reduced visual fidelity. Future work may integrate larger or more temporally expressive interpolation models to address these issues.

Failure modes 1: wrong object

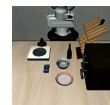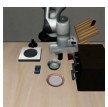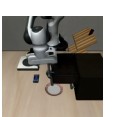     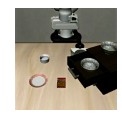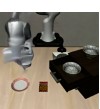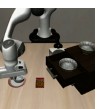

Prompt: open the middle drawer of the cabinet          Prompt: pick up the black bowl in the
                                                       top drawer of the wooden cabinet

Failure modes 2: blur and distortion

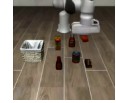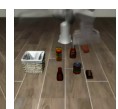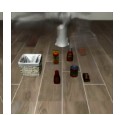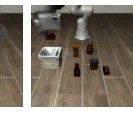     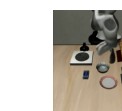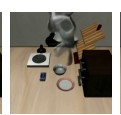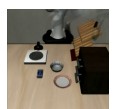

Prompt: pick up the butter and place it in the basket          Prompt: put the wine bottle on the top of the cabinet

Figure 11: Typical failure modes of the KeyWorld model.

## H    USE OF LLMs

The authors used LLMs to aid or polish paper writing, but all content has been carefully reviewed by the author. The authors used LLMs for literature retrieval and discovery, but all related works have been carefully reviewed and organized by the authors. The research ideation in this work was entirely completed by the author and does not involve the use of LLMs.

