# OpenReview forum: "KeyWorld: Key Frame Reasoning Enables Effective and Efficient World Models"
_ICLR.cc/2026/Conference — Submitted to ICLR 2026_

### Official Review · Reviewer_jUW1 · 2025-11-01

**Soundness:** 2
**Presentation:** 3
**Contribution:** 2
**Rating:** 4
**Confidence:** 4

**Summary:**

The paper introduces KeyWorld, a key-frame–based framework for robotic world models. It (i) selects motion-aware key frames via Ramer–Douglas–Peucker (RDP) over robot pose trajectories, (ii) uses a DiT/CogVideoX-style model to generate only those key frames, and (iii) interpolates the in-betweens with a lightweight CNN. On LIBERO, KeyWorld reports ~5.7× faster inference than full frame-to-frame generation with comparable or improved semantic/physical plausibility (incl. an object-level accuracy metric). Ablations vary key-frame density and compare RDP to uniform selection.

**Strengths:**

- Addresses an important computational bottleneck in video-based robot world models by reducing _unnecessary_ frame generation.

- The approach is simple and easy to reproduce: key-frame detection + generation + interpolation, with clear implementation details.

- Empirical results show substantial inference speedups (~5–8×) while maintaining comparable or in some cases better physical plausibility.

- The comparison against uniform frame selection highlights the value of motion-aware key-frame identification.

- Clear writing and diagrams that make the method easy to follow.

**Weaknesses:**

- Limited novelty: Key frame-based video generation is well-established (as acknowledged in Related Work). The main contribution is applying RDP to robotic trajectories to select such key frames, which is somewhat incremental.

- Evaluation limitations: Only evaluated on single-robot, fixed-camera scenarios and, most importantly, no comparison with other efficient world model alternatives (e.g., latent models, hierarchical models)

- Method limitations: (1)  Requires access to robot state information for RDP, limiting applicability to vision-only settings. (2) The interpolation model could fail for complex motions between key frames.

- Incomplete analysis:
---- No study of how performance degrades with interpolation distance.
---- Missing analysis of what makes a good key frame beyond motion transitions.
---- No discussion of failure modes

- Minor Issue: Inconsistent notation (x vs s for states)

**Questions:**

- Vision-only applicability: Is there a more general path for training this model without assuming access to poses? If not scalability of the method could be limited to robot-only (action-annotated) data. Could learned embeddings or motion cues substitute?

- Trade-off behavior:  Do you have curves or an analysis showing fidelity vs. key-frame sparsity? This would clarify how aggressively one can compress.

- Failure modes: What conditions break interpolation (fast contacts, occlusions, moving camera)? How would you go about mitigating this?

- Alternative salience signals: Did you try simple alternatives (velocity/jerk thresholds, embedding change magnitude, attention-based salience)? Any insights?

- Downstream planning: Have you tested whether KeyWorld outputs improve policy rollout or model-based planning performance? Currently, the evaluation focuses on video quality and plausibility, and demonstrating any downstream control benefit would significantly strengthen the robotics motivation and provide stronger support for some of the claims.

- Robustness / tuning: How sensitive is performance to the RDP tolerance? Any heuristic or automatic schedule for setting key-frame density?

---

> ### Author Response · Authors · 2025-11-21
> **Q1-3 Advantage of the motion-aware key frames**
>
> ### Q1 Tradeoff behavior.
> In our experiments, we find that the use of key frames does not introduce a trade-off between sparsity and fidelity in the usual sense. Instead, arranging key frames in this manner enhances the model’s focus on semantically important moments, reducing its burden of modeling continuous, smooth scenes between frames. This focused attention allows the model to better capture crucial actions and key objects, leading to improved performance overall (Table 2 in the paper). Figure 4 in the paper further analyzes the relationship between key-frame density and model performance. We observe that selecting roughly 10–20% of frames as key frames consistently achieves the strongest performance as well as significant acceleration.
>
> ### Q2 Comparing with efficient world models.
> Prior efficient world model or video generation approaches indeed improve computational efficiency through hierarchical modeling[1] or latent modeling [2], but they typically sacrifice spatial fidelity or temporal consistency
> Conversely, previous key-frame based world models usually focus on generating higher fidelity or longer videos but do not report reducing computation[3].
>
> In contrast, our work targets a different design point: achieving **both efficiency gains and improving physical plausibility**. This is enabled by a motion-aligned key-frame strategy tailored for robotic trajectories, which shifts computation toward physically meaningful states rather than uniformly dense predictions. Therefore, while related in spirit, existing efficient world models address orthogonal trade-offs and do not meet the combined goals of speed-up and physics-aware fidelity that we demonstrate.
>
> [1] Chen, Zigeng, et al. "Collaborative decoding makes visual auto-regressive modeling efficient." Proceedings of the Computer Vision and Pattern Recognition Conference. 2025.
>
> [2] Yuan, Zhihang, et al. "Dlfr-vae: Dynamic latent frame rate vae for video generation." Proceedings of the 33rd ACM International Conference on Multimedia. 2025.
>
> [3] Yang, Liudi, et al. "RoboEnvision: A Long-Horizon Video Generation Model for Multi-Task Robot Manipulation." arXiv preprint arXiv:2506.22007 (2025).
>
> ### Q3 Alternative salience signal.
> As robot states may not always be available, we additionally construct a pure vision-based key-frame detection alternative for video-only scenarios. We extract per-frame embeddings using a pretrained ResNet[4], apply Principal Component Analysis PCA to obtain low-dimensional “surrogate states,” and run the same RDP procedure to select key frames. We refer to this variant as **Visual**.
>
> This alternative makes our acceleration pipeline fully applicable in video-only settings. As shown in Table 1 below, Visual yields slightly lower video quality than our motion-aware key-frame strategy, indicating the advantage of using accurate robot states for aligning motion complexity. Nevertheless, Visual still produces high-quality videos and serves as a practical, efficient substitute when state information is unavailable. Figure 3 in the revised paper further illustrates this trade-off: the visual method may miss subtle motions, but remains a viable acceleration mechanism for standard video-only applications.
>
> Table 1: Video quality of different key frame arranging strategies. **Bold** refers to the best results.
> | Model | KeyWorld |  |  | Uniform |  |  | Visual |  |  |
> |:---:|:---:|:---:|:---:|:---:|:---:|:---:|:---:|:---:|:---:|
> | Metrics | PSNR | SSIM | Object-level accuracy | PSNR | SSIM | Object-level accuracy | PSNR | SSIM | Object-level accuracy |
> | LIBERO-10 | **19.07** |**0.8750** | **90\%** | 18.66 | 0.8400 | 88\% | 18.94 | 0.8738 | 87\% |
> | LIBERO-goal | **22.43** | **0.8838** | **90\%** | 21.97 | 0.8733 | 86\% | 22.11 | 0.8830 | 89\% |
> | LIBERO-object | **22.23** | 0.8801 | 62\% | 22.13 | 0.8844 | **70\%** | 22.01 | **0.8868** | 59\% |
> | LIBERO-spatial | **21.40** | 0.8694 | **33\%** | 21.32 | 0.8563 | 27\% | 21.21 | **0.8700** | 31\% |
>
> [4] He, Kaiming, et al. "Deep residual learning for image recognition." Proceedings of the IEEE conference on computer vision and pattern recognition. 2016.

---

> ### Author Response · Authors · 2025-11-21
> **Q4-7 Discussion on the details of KeyWorld model.**
>
> ### Q4 Optimal key frame density.
> The key frame density does not require manual tuning. Instead, we adopt an automatic key-frame selection strategy based on the elbow method in Appendix F of the updated paper. Specifically, we examine the residual curve produced by the RDP simplification process, analogous to selecting the number of clusters in K-means. As the key-frame density increases, the RDP residual decreases monotonically because the simplified trajectory better approximates the original one. We consistently observe a clear elbow point at which further increasing the number of key frames yields diminishing improvement. This point reflects the intrinsic motion complexity of the sequence and thus provides a principled criterion for determining the key-frame ratio.
> As reported in Appendix F in the revised paper, the elbow points across all LIBERO subsets fall around 10–15 key frames, which aligns with the empirical performance trends in Figure 5: configurations such as KeyWorld-10 (9 key frames) and KeyWorld-20 (17 key frames) achieve the best results. This demonstrates that the elbow-based automatic selection is consistent with the optimal performance region.
>
> Note: Due to the tokenizer design of the base CogVideoX-1.5 model, the number of key frames is constrained to the sequence 8N+1 (9, 17, 25...).
>
> ### Q5 Failure modes.
> We carefully examined the generated videos and summarized the main failure modes in Appendix G.
>
> The first type involves selecting or interacting with the incorrect object (e.g., manipulating the wrong drawer). This type of error is substantially reduced by KeyWorld, as reflected in its higher object-level accuracy. Such mistakes mainly arise from imperfect key-frame selection, and could be further mitigated by incorporating additional physical priors or lightweight post-training on the base model.
>
> The second type concerns occasional blur or distortion during fast motions. Since the key frames themselves remain high-quality, these artifacts primarily come from the interpolation stage when the predicted gap is not perfectly matched to the underlying motion speed. Stronger interpolation architectures could further reduce these issues, though we find them relatively infrequent in the evaluated benchmarks.
>
> ### Q6 Downstream planning tasks.
> Downstream planning is indeed an important application direction. Our work focuses on fast synthesis of high-quality, physically plausible videos that can serve as inputs for model-based planning or policy learning. In particular, the improved object-level accuracy of KeyWorld suggests that it may provide cleaner and more reliable visual rollouts, which are beneficial for downstream control. While a full integration with policy or planning frameworks is beyond the scope of this paper, we agree that this is a promising direction may conduct future research accordingly.
>
> ### Q7 Typos on notations.
> We carefully check and correct the notation for consistency and clarity. In the paper, the notation $x$ denotes video frames, $k$ denotes key-frame indices, and $s$ denotes robot states.
>
> Thanks again for your valuable suggestions! We will include the above discussions and improvements in our final version. Please let us know if you have any further concerns.

---

### Official Review · Reviewer_hi5p · 2025-11-01

**Soundness:** 2
**Presentation:** 3
**Contribution:** 2
**Rating:** 4
**Confidence:** 2

**Summary:**

The paper introduces a keyframe-centric framework for text-conditioned robotic world models. It first extracts motion-aware key frames from robot pose trajectories, trains a DiT-based generator (finetuned from CogVideoX) to synthesize semantically critical frames, and then uses a lightweight CNN gap estimator plus FILM interpolator to inpaint intermediate frames. On LIBERO dataset, KeyWorld cuts end-to-end inference time to <25% of frame-to-frame generation while maintaining PSNR/SSIM.

**Strengths:**

- The problem motivation is strong and the work has clear decomposition. The work improves redundancy limits in per-frame rollouts and cleanly splits generation into key-frame reasoning + lightweight interpolation.

- The total latency drops to 17–18% of baseline across splits, with comparable PSNR/SSIM and large gains in object-level correctness on some sets.

**Weaknesses:**

- Authors note that large pose differences make intermediate motion “difficult to model,” yet the interpolator is a lightweight CNN-style module, potentially brittle under long horizons or contact-rich transitions.

- The approach explicitly targets scenes where observations are captured by a fixed camera, limiting applicability to moving setups.

- Results are only on LIBERO variants, generalization to other robotic datasets or real-world videos is unclear.

- As a video world model paper, no qualitative video results are provided. It's hard to evaluate the performance without watching the videos.

- The paper claims that the proposed method improves "physical validity" of the generated videos. However, the metrics contain only PSNR/SSIM and object-level correctness, these do not directly assess physics/contacts or human preference.

**Questions:**

- How would the pipeline adapt to camera motion. By estimating and compensating ego-motion before RDP selection or conditioning the generator on camera trajectory?

- Could the authors validate on additional benchmarks or real robot captures (varying viewpoints and lighting) to test generalization beyond LIBERO?

- Could the author provide qualitative video results?

- Could the author conduct human preference studies to substantiate “higher physical plausibility” claims?

---

> ### Author Response · Authors · 2025-11-21
>
> ### Q1 Selection of the light-weight CNN model
> We choose a lightweight CNN interpolator to ensure fast inference. To reduce the difficulty of modeling large pose changes, our method decomposes the overall motion into multiple segments that are simple (approximately straight movements rather than curves) in robot pose space. Under this design, even a small CNN has sufficient capacity to generate accurate intermediate frames while maintaining high efficiency.
>
> ### Q2 Tackling moving camera and views
> Our current focus is on third-person robotic demonstrations with largely fixed viewpoints and limited occlusion. This is the standard setting used in existing robotic world-model datasets and benchmarks (e.g., JEPA [1], WorldVLA [2], RT-1 [3]), as well as in downstream policy-learning environments (e.g., RoboScape [4], RoboEnvision [5]). Under these conditions, robot kinematics remain visually observable, so RDP-selected key frames stay well aligned with salient pixel-space transitions.
>
> We agree that moving-camera, first-person, and multi-view setups are important and arise in domains such as autonomous driving. These settings introduce rapid background changes and view-dependent appearance variations, which pose additional challenges for interpolation quality. Addressing them likely requires stronger, view-aware interpolation components (e.g., diffusion-based or geometry-informed models). Extending our framework to such settings is a valuable direction for future work, and we appreciate the reviewer for bringing up this point.
>
> [1] Assran, Mido, et al. "V-jepa 2: Self-supervised video models enable understanding, prediction and planning." arXiv preprint arXiv:2506.09985 (2025).
>
> [2] Cen, Jun, et al. "WorldVLA: Towards Autoregressive Action World Model." arXiv preprint arXiv:2506.21539 (2025).
>
> [3] Brohan, Anthony, et al. "Rt-1: Robotics transformer for real-world control at scale." arXiv preprint arXiv:2212.06817 (2022).
>
> [4] Shang, Yu, et al. "RoboScape: Physics-informed Embodied World Model." arXiv preprint arXiv:2506.23135 (2025).
>
> [5] Yang, Liudi, et al. "RoboEnvision: A Long-Horizon Video Generation Model for Multi-Task Robot Manipulation." arXiv preprint arXiv:2506.22007 (2025).
>
> ### Q3 Transferring to more benchmarks.
> Due to limited computing resources, we are currently unable to fully train and evaluate our model on additional benchmarks, especially those involving moving or multi-view cameras. Nevertheless, motion-based key-frame selection itself is highly transferable, and we evaluate it on real-world robotic examples of the AgiBot World Challenge dataset[6], which features different scenes and object configurations. As shown in Appendix D in the revised paper, the extracted key frames align well with major semantic phases of the manipulation process (e.g., reaching, grasping, lifting). This indicates that our decomposition method is not tied to a specific dataset and can generalize to novel environments.
>
> [6] Bu, Qingwen, et al. "Agibot world colosseo: A large-scale manipulation platform for scalable and intelligent embodied systems." arXiv preprint arXiv:2503.06669 (2025).
>
> ### Q4 Visualization results.
> We updated the visualization results in the GitHub link, which may assist your evaluation of the performance of our model.
>
> [https://anonymous.4open.science/r/Keyworld-E43D/visualization/readme.md](https://anonymous.4open.science/r/Keyworld-E43D/visualization/readme.md)
>
> ### Q5 Human-centric metrics.
> While we do not conduct a separate human study, our object-level accuracy metric is fully based on human judgment: annotators verify whether the key objects are correctly positioned and interacting as expected. Mistakes in object placement or motion lead to visually obvious distortions, which are inherently inconsistent with physical laws. Therefore, high object-level accuracy implicitly reflects human perception of physical plausibility in the generated videos.
>
> Thanks again for your valuable suggestions! We will include the above discussions and improvements in our final version. Please let us know if you have any further concerns.

---

### Official Review · Reviewer_AKto · 2025-11-02

**Soundness:** 2
**Presentation:** 3
**Contribution:** 2
**Rating:** 4
**Confidence:** 3

**Summary:**

This paper describes a method KeyWorld. Instead of generating motion predictions frame by frame, the authors observe that there are certain frames that are "key" for prediction, while other intermediate frames can be inferenced by interpolating the key frames.

The method has two steps:
1. a key frame generation model trained on key frames extracted using robot pose vectors, these frames are selected to be sensitive to pose instead of uniform.
2. reconstruction of the whole video by interpolating the frames using the FILM method.

**Strengths:**

The paper choose to use key frames to resolve the key speed limitation for robotic WM deployment. The time needed for generation is only about 15-20% of the total time (or 5 times speedup) but with improved video generation quality.

**Weaknesses:**

**Reliance on pose data**: to extract the key frames data, this method requires the RDP algorithm with robot pose data, which is an extra requirement compared with WM training with mainly video only data, or prior key frame aware work that utilized task descriptions.

**Several task specific assumptions": this paper makes a few assumptions, which are valid to the tasks studied here, but the assumptions may be too strong.
  - The gap estimator assumes "robot generally moves at a relatively stable speed". In the experiment of this paper, these estimators seems to be relatively ok, as termed "acceptable" in appendix E.
  - The method assumes the kinematic changes are good estimators for key frames. This assumption work well with the experimental dataset used.  However, this assumption may need more proper discussion on its potential limitation, and maybe experiments on more domains to confirm this.

**Questions:**

- The authors try to approach the problem of "real-time" robotic control, but the method still requires minutes to do the prediction. Further, to generate next frames, the model needs to generate the next key frame first, which may creates another obstacle for reaching real-time.
- The robot pose vectors are currently key to the method. What is a proposed way to overcome this. If the "smooth" frame transition is true, could we use that as a way to generate good key frame data instead.

---

> ### Author Response · Authors · 2025-11-21
>
> ### Q1 Reliance on pose data:
> As robot states may not always be available, we additionally construct a pure vision-based key-frame detection alternative for video-only scenarios. We extract per-frame embeddings using a pretrained ResNet[1], apply Principal Component Analysis (PCA) to obtain low-dimensional “surrogate states,” and run the same RDP procedure to select key frames. We refer to this variant as **Visual**.
>
> This alternative makes our acceleration pipeline fully applicable in video-only settings. As shown in Table 1 below, Visual yields slightly lower video quality than our motion-aware key-frame strategy, indicating the advantage of using accurate robot states for aligning motion complexity. Nevertheless, Visual still produces high-quality videos and serves as a practical, efficient substitute when state information is unavailable. Figure 3 in the revised paper further illustrates this trade-off: the visual method may miss subtle motions such as the end effectors, but remains a viable acceleration mechanism for standard video-only applications.
>
> Table 1: Video quality of different key frame arranging strategies. **Bold** refers to the best results.
> | Model | KeyWorld |  |  | Uniform |  |  | Visual |  |  |
> |:---:|:---:|:---:|:---:|:---:|:---:|:---:|:---:|:---:|:---:|
> | Metrics | PSNR | SSIM | Object-level accuracy | PSNR | SSIM | Object-level accuracy | PSNR | SSIM | Object-level accuracy |
> | LIBERO-10 | **19.07** |**0.8750** | **90\%** | 18.66 | 0.8400 | 88\% | 18.94 | 0.8738 | 87\% |
> | LIBERO-goal | **22.43** | **0.8838** | **90\%** | 21.97 | 0.8733 | 86\% | 22.11 | 0.8830 | 89\% |
> | LIBERO-object | **22.23** | 0.8801 | 62\% | 22.13 | 0.8844 | **70\%** | 22.01 | **0.8868** | 59\% |
> | LIBERO-spatial | **21.40** | 0.8694 | **33\%** | 21.32 | 0.8563 | 27\% | 21.21 | **0.8700** | 31\% |
>
> [1] He, Kaiming, et al. "Deep residual learning for image recognition." Proceedings of the IEEE conference on computer vision and pattern recognition. 2016.
>
> ### Q2: Regarding the task-specific assumptions:
> We agree that our method relies on two assumptions: (1) the robot moves at a roughly stable speed for gap estimation, and (2) kinematic changes serve as reliable indicators for key frames. We examined both assumptions on two robotic manipulation datasets: the LIBERO dataset used for training and evaluation, and a real-world robot-operation dataset (Agibot World Challenge[2]). The results support the validity of these assumptions.
>
> For the assumption of relatively steady robot motion, we analyzed the state and speed trajectories across all tasks. A K-means clustering and t-SNE visualization of the speed vectors (Appendix E of the revised paper) shows that the motions form several well-separated clusters, indicating that the robot follows a small set of motion primitives with relatively consistent speed patterns. This supports the stability assumption behind our gap estimator and suggests that it remains applicable across similar robotic manipulation domains.
>
> To further verify that kinematic changes correlate with semantic key frames, we include an example of detected key frames of the Agibot World Challenge dataset[2] in Appendix D of the revised paper. The motion-derived key frames align well with the semantic stages of the task, such as reaching, grasping, and lifting the object. This demonstrates that kinematic turning points are indeed strong proxies for meaningful visual and task-relevant changes across multiple datasets.
>
>
> [2] Bu, Qingwen, et al. "Agibot world colosseo: A large-scale manipulation platform for scalable and intelligent embodied systems." arXiv preprint arXiv:2503.06669 (2025).
>
> ### Q3: Real-time robotic operation
> Our goal is to significantly accelerate video synthesis within robotic world models, which is critical for data generation and simulation-heavy workflows. While our current system still does not achieve real-time generation speed, the proposed key-frame mechanism offers substantial speedups over full-frame generation. We have revised the manuscript to clarify this scope and avoid implying real-time execution, focusing on acceleration for data synthesis and model training.
>
> Thanks again for your valuable suggestions! We will include the above discussions and improvements in our final version. Please let us know if you have any further concerns.

---

### Official Review · Reviewer_EYHT · 2025-11-04

**Soundness:** 3
**Presentation:** 3
**Contribution:** 2
**Rating:** 4
**Confidence:** 3

**Summary:**

Summary
This paper proposes a key-frame reasoning + lightweight interpolation paradigm for world-model video generation, designed to efficiently produce physically consistent videos and robot motion trajectories. The overall generation pipeline consists of four main components:
	1.	Key-frame selection (offline).
Starting from the robot’s state sequence (poses or joint angles), the method applies the Ramer–Douglas–Peucker (RDP) algorithm to extract key-frame indices at kinematic turning points, which represent only a small fraction of the total frames.
	2.	Key-frame generation (heavy model).
Conditioned on the first-frame image and a textual instruction, the system uses a diffusion/DiT model (based on CogVideoX) to synthesize or edit only the selected key frames. This concentrates computation on a small set of critical frames, significantly reducing latency.
	3.	Gap prediction (lightweight network).
By encoding the visual differences between adjacent key frames, a small CNN/MLP predicts the number of intermediate frames (the gap) that should be inserted between them.
	4.	Interpolation and reconstruction (lightweight module).
Using FILM or similar lightweight interpolation modules, the system generates the intermediate frames based on the predicted gap and reconstructs the complete video or motion sequence.

⸻

Contributions
	1.	Motion-driven key-frame paradigm.
Key frames are selected from robot states rather than pixels or uniform sampling, aligning with physical semantic transitions such as grasping, lifting, and placing, thereby reducing redundant computation.
	2.	Two-stage generation design.
The computationally expensive diffusion/DiT model is applied only to key frames, while the remaining frames are generated through lightweight interpolation. This results in approximately linear scaling between compute cost and the proportion of key frames.
	3.	Minimal gap-prediction head.
A small model estimates the temporal span between key frames, improving interpolation stability and temporal consistency.
	4.	Balanced efficiency–quality trade-off.
On standard robotic benchmarks, the proposed method achieves multi-fold speedups over frame-by-frame generation while maintaining or improving image quality and object/action consistency. Extensive ablation studies analyze the effects of key-frame density and sampling strategies.

**Strengths:**

Originality
	1.	Motion-driven key-frame paradigm.
Unlike common approaches that use uniform key-frame sampling or language-step alignment, this work selects key frames directly from the robot’s state sequence using the Ramer–Douglas–Peucker (RDP) algorithm at kinematic turning points, and concentrates heavy computation on these frames. This design forms a two-stage pipeline combining key-frame reasoning with lightweight interpolation. For world-model video generation, it redefines the notion of “key frames” from semantic or time-uniform choices to physically aligned, state-driven selections—a clearly novel perspective.
	2.	Compute–key-frame linear scaling.
The inference cost scales approximately linearly with the proportion of selected key frames, making computational expense a tunable parameter. Such an explicit and controllable trade-off between cost and performance is rarely seen in comparable generation methods.

⸻

Quality
	1.	Significant speedup.
On benchmark tasks, the overall inference latency is reduced to roughly 17–25% of that of full frame-by-frame generation. Depending on the key-frame density, the reported acceleration ranges from approximately 3× to 8×, representing a substantial efficiency gain.
	2.	Maintained or improved visual and physical consistency.
Despite handling far fewer frames with the heavy generator, the method achieves PSNR and SSIM values comparable to or better than strong baselines. Moreover, object-level accuracy on the KeyWorld benchmark surpasses uniform key-frame selection, demonstrating that focusing computation on physically meaningful turning points helps capture critical events such as contact, grasping, and lifting.

⸻

Clarity
	1.	Clear modular pipeline and simple interfaces.
The approach is transparently structured as four modular components—key-frame selection (offline), key-frame generation (DiT), gap prediction (MLP), and interpolation (FILM). Each stage has well-defined inputs, outputs, and dependencies, which greatly facilitates reproducibility and extension.

**Weaknesses:**

Weaknesses (Specific and Actionable)
	1.	Key-frame idea is not entirely new, but rather a state-anchored variant.
The concept of sparse generation plus interpolation has prior art in both video generation and robotic visualization—such as uniform sampling, language-step decomposition, and key-frame selection based on optical flow or action boundaries. The main novelty here lies in using robot states with the Ramer–Douglas–Peucker (RDP) algorithm for motion alignment. However, the paper does not include direct comparisons against language-step or semantic subgoal baselines, nor against purely visual action-boundary detection methods.
	2.	Dependence on accurate robot states limits applicability to video-only settings.
The method assumes access to precise pose or joint-state information. In many real-world, third-person videos (e.g., YouTube clips, handheld viewpoints, or moving cameras), such state data are unavailable. Without additional estimation mechanisms, the RDP-based key-frame selection cannot be directly applied.
	3.	Robustness under moving cameras, multi-view setups, and heavy occlusion is unverified.
Kinematic turning points in robot state space do not necessarily correspond to visually salient changes in pixel space when the camera is moving or when occlusions occur. Consequently, RDP-selected key frames may become misaligned with visually important moments, leading to large and potentially unstable interpolation gaps between key frames.

⸻

**Questions:**

Questions
	1.	Alternatives for video-only settings.
The current key-frame selection depends on robot states (poses and joints). What if only videos are available, such as those with moving cameras or third-party footage? Could you provide a substitute key-frame selector based on visual motion estimation, optical flow, or action-boundary detection, and report its performance gap against the state + RDP approach in terms of PSNR/SSIM, object-level consistency, and latency?
	2.	Cross-paradigm comparison: language steps vs. semantic subgoals.
Under the same training budget and resolution, can you include a direct comparison against language-step decomposition or semantic-subgoal key-frame selection? It would be helpful to quantify how the three paradigms—state-aligned, language-aligned, and visual-boundary—differ in object consistency, contact-event recall, and overall latency.
	3.	Generalization and robustness of key-frame selection.
Do the RDP threshold and the resulting key-frame ratio require manual tuning across different tasks and camera setups? Could you propose an adaptive scheme based on uncertainty or motion magnitude for threshold or ratio selection, and include benefit curves showing its impact on performance?

---

> ### Author Response · Authors · 2025-11-21
> **Q1: Comparison of different key-frame selection strategies.**
>
> ### Q1: Comparison of different key-frame selection strategies.
> ### A1.1: Comparison with vision-based key-frame detection.
> We construct a pure vision-based key-frame detection strategy, which is denoted as **Visual** below. We extract per-frame embeddings using a pretrained ResNet[1], apply Principal Component Analysis (PCA) to obtain low-dimensional "surrogate states", and run the same RDP procedure to identify key frames. This baseline does not require access to states and provides an alternative application for our acceleration pipeline to most video datasets.
>
> As shown in the Table below, Visual consistently produces lower video quality than our motion-aware variant, confirming the benefit of using true robot states for trajectory alignment. A more detailed example of these key-frame strategies is included in Figure 3 in the revised paper: the visual key-frame detector fails to capture subtle but crucial motions, such as the precise end-effector movement when releasing a can. This leads to desynchronized or incorrect temporal pacing relative to the ground truth.
>
> Table 1: Video quality of different key frame arranging strategies. **Bold** refers to the best results.
> | Model | KeyWorld |  |  | Uniform |  |  | Visual |  |  |
> |:---:|:---:|:---:|:---:|:---:|:---:|:---:|:---:|:---:|:---:|
> | Metrics | PSNR | SSIM | Object-level accuracy | PSNR | SSIM | Object-level accuracy | PSNR | SSIM | Object-level accuracy |
> | LIBERO-10 | **19.07** |**0.8750** | **90\%** | 18.66 | 0.8400 | 88\% | 18.94 | 0.8738 | 87\% |
> | LIBERO-goal | **22.43** | **0.8838** | **90\%** | 21.97 | 0.8733 | 86\% | 22.11 | 0.8830 | 89\% |
> | LIBERO-object | **22.23** | 0.8801 | 62\% | 22.13 | 0.8844 | **70\%** | 22.01 | **0.8868** | 59\% |
> | LIBERO-spatial | **21.40** | 0.8694 | **33\%** | 21.32 | 0.8563 | 27\% | 21.21 | **0.8700** | 31\% |
>
> [1] He, Kaiming, et al. "Deep residual learning for image recognition." Proceedings of the IEEE conference on computer vision and pattern recognition. 2016.
>
> ### A1.2: Comparison with language-based decomposition.
> For language-step decomposition, the main practical limitation is the lack of publicly available datasets with fine-grained language–video alignment. Natural language descriptions typically summarize the entire trajectory, and their decomposed sub-instructions are not temporally grounded to specific frames. As a result, training a language-conditioned key-frame generator is currently infeasible in the public domain; prior works using this paradigm rely on private datasets[2].
>
> A second difficulty is that language-based decomposition typically produces much sparser step boundaries than video or state-based segmentation. For tasks such as “Turn on the stove” or “Put the alphabet soup into the basket”, it is challenging to obtain more than 5 meaningful linguistic sub-steps, which is significantly below the empirically near-optimal key-frame density (~10–15) discussed in Q3. The resulting sparsity substantially increases the difficulty of interpolation and often leads to large temporal gaps.
>
> We conducted a small-scale qualitative comparison of the language-based decomposition: we decomposed tasks using Qwen2.5-VL-72B, manually aligned each sub-instruction to its closest ground-truth frame, and applied the same interpolator used in our method. As shown in the visualization examples [https://anonymous.4open.science/r/Keyworld-E43D/language-example/readme.md](https://anonymous.4open.science/r/Keyworld-E43D/language-example/readme.md), the sparse language-selected key frames produce noticeably blurrier and less stable reconstructions, confirming the challenge of relying solely on linguistic boundaries for dense video prediction.

---

> ### Author Response · Authors · 2025-11-21
> **Q2 (moving camera) and Q3 (optimal key frame density)**
>
> ### Q2 Tackling moving camera and views
> Our current focus is on third-person robotic demonstrations with largely fixed viewpoints and limited occlusion. This is the standard setting used in existing robotic world-model datasets and benchmarks (e.g., JEPA [3], WorldVLA [4], RT-1 [5]), as well as in downstream policy-learning environments (e.g., RoboScape [6], RoboEnvision [2]). Under these conditions, robot kinematics remain visually observable and occlusion is mild, so RDP-selected key frames stay well aligned with salient pixel-space transitions.
>
> We agree that moving-camera, first-person, and multi-view setups are important and arise in domains such as autonomous driving. These settings introduce rapid background changes and view-dependent appearance variations, which pose additional challenges for interpolation quality. Addressing them likely requires stronger, view-aware interpolation components (e.g., diffusion-based or geometry-informed models). Extending our framework to such settings is a valuable direction for future work, and we appreciate the reviewer for bringing up this point.
>
> [2] Yang, Liudi, et al. "RoboEnvision: A Long-Horizon Video Generation Model for Multi-Task Robot Manipulation." arXiv preprint arXiv:2506.22007 (2025).
>
> [3] Assran, Mido, et al. "V-jepa 2: Self-supervised video models enable understanding, prediction and planning." arXiv preprint arXiv:2506.09985 (2025).
>
> [4] Cen, Jun, et al. "WorldVLA: Towards Autoregressive Action World Model." arXiv preprint arXiv:2506.21539 (2025).
>
> [5] Brohan, Anthony, et al. "Rt-1: Robotics transformer for real-world control at scale." arXiv preprint arXiv:2212.06817 (2022).
>
> [6] Shang, Yu, et al. "RoboScape: Physics-informed Embodied World Model." arXiv preprint arXiv:2506.23135 (2025).
>
>
> ### Q3: Selecting the density of the key-frames
> The key frame density does not require manual tuning. Instead, we adopt an automatic key-frame selection strategy based on the elbow method in Appendix F of the revised paper. Specifically, we examine the residual curve generated by the RDP simplification process, which is analogous to selecting the number of clusters in K-means. As the key-frame density increases, the RDP residual decreases monotonically because the simplified trajectory better approximates the original one. We consistently observe a clear elbow point at which further increasing the number of key frames yields diminishing improvement. This point reflects the intrinsic motion complexity of the sequence and thus provides a principled criterion for determining the key-frame ratio.
> As reported in Appendix F, the elbow points across all LIBERO subsets fall around 10–15 key frames, which aligns with the empirical performance trends in Figure 5: configurations such as KeyWorld-10 (9 key frames) and KeyWorld-20 (17 key frames) achieve the best results. This demonstrates that the elbow-based automatic selection is consistent with the optimal performance region.
>
> Note: Due to the tokenizer design of the base CogVideoX-1.5 model, the number of key frames is constrained to the sequence 8N+1 (9, 17, 25...).
>
> Thanks again for your valuable suggestions! We will include the above discussions and improvements in our final version. Please let us know if you have any further concerns.

---

### Meta-Review · Area_Chair_mmXv · 2025-12-31

**Summary:**

The major concern raised by the reviewers lies on that KeyWorld requires accurate robot state information for the proposed RDP algorithm to select key frames – limiting its application in video-only settings where real-world data is more readily available. Additionally, the reviewers also expressed concerns about the limited evaluation scope, which focuses exclusively on the LIBERO dataset under relatively constrained settings, as well as questions regarding the novelty of the approach compared to well-established key-frame-based video generation methods. As these core issues remain largely unresolved during the rebuttal phase, I would recommend rejection at this point and encourage the authors to further refine and strengthen the work.

**Reviewer Concerns:**

During the rebuttal, the authors provided clarifications on the fixed-camera settings, key-frame density selection criterion, and failure modes, which helped address some questions raised. However, the core limitation of KeyWorld under video-only settings remains. Though the authors offered a simple vision-based variant using PCA features extracted from ResNet for key-frame detection, the reported results indicate this alternative consistently produces lower video quality, thus questioning KeyWorld's applicability under video-only settings. Furthermore, limited evidence beyond the LIBERO dataset or demonstrations on downstream applications is provided. I believe the main concerns are still outstanding.

**Reviewer Scores:**

Although the author provided clarifications regarding the fixed-camera setting and key-frame density selection criterion, Reviewer EYHT is likely to maintain the score, as the given vision-based alternative consistently produces lower video quality, confirming the key limitation of the proposed method.

Reviewer AKto (4) may also maintain the score due to the method’s reliance on pose data.

Reviewer hi5p (4) may maintain the score as well, as evidence beyond the LIBERO dataset remains limited, despite the authors’ clarifications on method and evaluation choices.

Reviewer jUW1(4) may maintain the score because the weaknesses of novelty, evaluation, and method limitations remain unaddressed.

Overall, the limitations raised by the reviewers have not been sufficiently addressed in the rebuttal, and I would expect the reviewers to continue leaning towards rejection.

---

### Decision · Program_Chairs · 2026-01-26

Reject